# Endowing Acceptable Mechanical Properties of Segregated Conductive Polymer Composites with Enhanced Filler-Matrix Interfacial Interactions by Incorporating High Specific Surface Area Nanosized Carbon Black

**DOI:** 10.3390/nano11082074

**Published:** 2021-08-16

**Authors:** Huibin Cheng, Xiaoli Sun, Baoquan Huang, Liren Xiao, Qinghua Chen, Changlin Cao, Qingrong Qian

**Affiliations:** 1College of Environmental Science and Engineering, Fujian Normal University, Fuzhou 350007, China; QBX20190099@yjs.fjnu.edu.cn (H.C.); sunxiaoli@fjnu.edu.cn (X.S.); qbh811@fjnu.edu.cn (B.H.); cqhuar@fjnu.edu.cn (Q.C.); 2Engineering Research Center of Polymer Green Recycling of Ministry of Education, Fuzhou 350007, China; xlr1966@fjnu.edu.cn; 3Fujian Key Laboratory of Pollution Control & Resource Reuse, Fuzhou 350007, China

**Keywords:** interfacial interaction, conducting carbon black network, mechanical property, electromagnetic interference shielding

## Abstract

Tuning the high properties of segregated conductive polymer materials (CPCs) by incorporating nanoscale carbon fillers has drawn increasing attention in the industry and academy fields, although weak interfacial interaction of matrix-filler is a daunting challenge for high-loading CPCs. Herein, we present a facile and efficient strategy for preparing the segregated conducting ultra-high molecular weight polyethylene (UHMWPE)-based composites with acceptable mechanical properties. The interfacial interactions, mechanical properties, electrical properties and electromagnetic interference (EMI) shielding effectiveness (SE) of the UHMWPE/conducting carbon black (CCB) composites were investigated. The morphological and Raman mapping results showed that UHMWPE/high specific surface area CCB (h-CCB) composites demonstrate an obviously interfacial transition layer and strongly interfacial adhesion, as compared to UHMWPE/low specific surface area CCB (l-CCB) composites. Consequently, the high-loading UHMWPE/h-CCB composite (beyond 10 wt% CCB dosage) exhibits higher strength and elongation at break than the UHMWPE/l-CCB composite. Moreover, due to the formation of a densely stacked h-CCB network under the enhanced filler-matrix interfacial interactions, UHMWPE/h-CCB composite possesses a higher EMI SE than those of UHMWPE/l-CCB composites. The electrical conductivity and EMI SE value of the UHMWPE/h-CCB composite increase sharply with the increasing content of h-CCB. The EMI SE of UHMWPE/h-CCB composite with 10 wt% h-CCB is 22.3 dB at X-band, as four times that of the UHMWPE/l-CCB composite with same l-CCB dosage (5.6 dB). This work will help to manufacture a low-cost and high-performance EMI shielding material for modern electronic systems.

## 1. Introduction

With the rapid advancement in electronic information technology, electronic equipment and communication base stations have become an indispensable part of people’s daily lives. Meanwhile, this originated from man-made electronic devices accompanied by high-energy electromagnetic wave radiation pollution and electromagnetic interference have become critical problems that need to be addressed [1,2]. Current electromagnetic interference (EMI) shielding materials mainly include metal and their alloy materials, but these materials have many shortcomings, such as high density, high cost, low corrosions and a complicated manufacturing process [3]. Therefore, the electrically conductive polymer composites (CPCs) containing conductive fillers were identified as one of the excellent candidates for EMI shielding materials due to their excellent merits, such as low density, regulable electrical conductivity and corrosion resistance, etc. [4,5,6,7]. To effectively reduce the impact of electromagnetic wave radiation on interfering with the equipment and human health, especially, people paid widespread attention to the design and fabrication of lightweight and highly electrical conducting carbon fillers filled CPCs. In recent years, to obtain higher EMI shielding effectiveness (SE) value, CPCs must require sufficient conductive loss and interface polarized loss. Additionally, a reasonable layout of the conductive network is necessary [8,9,10]. The construction of stable, efficient and integral conductive paths can be helpful for conductive loss. Unfortunately, highly efficient conductive pathways always require high content filler loading, which inevitably affects the mechanical properties and processibility of materials [11,12]. Thus, how to further obtain an efficient and dense conductive network at low content loading for EMI shielding polymer composites still remains a daunting challenge [13].

The segregated conductive pathway has proven to outperform in enlarging conductive loss mechanism, particularly for lightweight polymer-based EMI shielding materials with segregated structure, multi-element composite structure, laminated structure and multi-interface structure [13,14,15,16,17]. Unfortunately, the segregated conductive pathways may cause low enhancement efficiency of mechanical properties. The choice of the polymer matrices and electrically conductive fillers, and the interface microstructure of polymer-filler are very important for solving undesirable mechanical properties and manufacturing high-performance CPCs. Currently, tremendous efforts in overcoming the disperse and weak interfacial interaction of nanocarbon filler, such as carbon nanotube (CNT) [18,19,20,21,22], graphite (G) [23], carbon black [21,24] and graphene nanoplatelets (GNPs) [25,26], etc. Numerous works have attempted to improve the interface interaction of CPCs by the synergistic effect of different dimensional filler and surface modification of filler [27,28]. Li [23] et al. reported hybrid graphite/carbon black (CB)/ultrahigh molecular weight polyethylene (UHMWPE) composite exhibited a high electrical conductivity and superb EMI SE. The mechanical enhancement is attributed to the synergistic effect of the nanosized CB particles and large-aspect-ratio graphite platelets, providing a stronger interfacial adhesion between UHMWPE domains. Nah, et al. [29] developed a highly scalable EMI shielding material comprising nanostructured carbon black and poly (methyl methacrylate) (PMMA) nanocomposites via solution mixing, followed by the compression molding method. The obtained PMMA nanocomposite with 10 wt% filler loading possessed a superior EMI SE value of ~28 dB at X band. Moreover, inspired by the microstructure of the bio-structural materials [30,31,32], people learn from nacre to construct unique structural polymer composites—that is, design the orderly morphology of multi-layered structural polymer matrix and the alignment of inorganic fillers. The thermal conductivity, electrical properties and mechanical properties of polymer/filler nanocomposites were improved via interface engineering [33,34]. However, among these methods in enhancing the filler-matrix, interfacial interactions is relatively complicated for the practical production process. Thus, for CPCs with strong interface adhesion and small defects, whether they can simply employ low-cost and high-electrically conductive nanostructured carbon fillers for developing EMI shielding materials has become a very perspective research hotspot [35,36].

In this work, UHMWPE with unique rheological property serves as a matrix of segregated conductive polymer composites, and nanosized carbon black (CCB) with high and low specific surface area acts as conductive fillers. A comparative study of CPCs with a dense high specific surface area conductive carbon black (h-CCB) packing network and a low specific surface area conductive carbon black (l-CCB) loose network based on an UHMWPE matrix was fabricated through mechanical mixing and compression molding methods. We found that h-CCB particles with the multi-nanopores and intensive polar interaction can not only be used as a conducting agent to enhance the electrical conductivity of UHMWPE, but also as transition layer to improve the filler-matrix interfacial interactions, which in turn effectively suppress interfacial defects between the adjacent UHMWPE and h-CCB domains, and greatly form a highly efficient dense conductive network in the composite. The achieved high-loading UHMWPE-based CPCs exhibited a significant improvement of EMI shielding performance and mechanical properties. Furthermore, to meet the requirement of actual applications, the influence of h-CCB and l-CCB content on phase morphology, interfacial bonding mechanism, thermal properties and EMI SE of high loading UHMWPE/CCB composites beyond 10 wt% CCB dosage were investigated comprehensively.

## 2. Materials and Methods

### 2.1. Materials

CCB (F900A) with a high specific surface area (h-CCB) of 380 m^2^/g were purchased from Tianjin Ebory Chemical Co., Ltd., (Tianjin, China). CCB (Orion N990) with a low specific surface area (l-CCB) of 8 m^2^/g was obtained from Shenzhen Securities Yoshida Chemical Co., Ltd., (Shenzhen, China). Comparison of the specific surface area and pore size distribution of two kinds of CCB nanoparticles were given in Appendix A and Appendix A. UHMWPE, under the trade of SLL-2, with an average molecular weight of 3.00 × 10^6^ g/mol, was supplied by Shanghai Lianle Chemical Industry Science and Technology Co., Ltd., (Shanghai, China). All raw materials were used as received without further modification.

### 2.2. Preparation of UHMWPE/CCB Composites

The preparation schematic illustration of segregated UHMWPE/CCB composites is presented in Figure 1. At first, h-CCB particles and UHMWPE powder were dried in a vacuum oven at 80 °C for 6 h. Then, UHMWPE powder and high specific surface area h-CCB particles with compositions of 99.5/0.5, 99/1, 97/3, 95/5, 93/7, 90/10 and 85/15 were premixed by adopting the crusher high-speed mechanical mixing (800Y, Yongkang Platinum Ou Hardware Products Co., Ltd., Zhejiang, China) at a rotation speed of 34,000 r/min for 50 s, obtaining the high specific surface area CCB-coated UHMWPE mixtures. Afterward, the obtained compounds were compression molded at 200 °C for 20 min on a flat vulcanizing machine (ZG-80 T, Guangdong Zhenggong Electromechanical Equipment Technology Co., Ltd., Guangdong, China), followed by cold compression molded to room temperature at a pressure of 17 MPa. The final obtained specimens were denoted as UHMWPE/h-CCB_x_ composites, in which x means that the weight fraction of h-CCB nanoparticles. For comparison, l-CCB nanoparticles modified UHMWPE composites with the same weight ratios were prepared via high-speed mechanical mixing and compression molding. The samples were remarked as UHMWPE/l-CCB_x_ composites.

### 2.3. Characterization

The morphologies of CCB nanoparticles and composites were investigated using a cold field emission scanning electron microscope (FE-SEM, Regulus 8100, Hitachi, Japan) at an accelerating voltage of 10 kV. All samples were fractured in liquid nitrogen and sputter-coated with gold before observation. 

The electrical conductivity was measured by a high-resistance meter (ZC-90G, Shanghai Taiou Electronics Co., Ltd., Shanghai, China) higher than 10^−5^ S/m, four-point probes resistivity measurement (RTS-9, Guangzhou Four Probe Technology Co., Ltd., Guangzhou, China) below 10^−5^ S/m. The electromagnetic interference shielding performance of the samples depended on scattering parameters, which corresponded to the reflection and transmission of transverse electromagnetic waves, was performed by a Vector network analysis (N5244A, Agilent Technologies, SCC, CA, USA) with the wave-guide method at X-band frequency range (8.2–12.4 GHz) according to ASTM D5568-08, the resultant scattering parameters were used to calculate EMI SE values.

The sheets were cut into dumbbell-shaped specimens for the evaluation of the mechanical properties using a universal testing machine (CMT4104, Shenzhen Sans Material Inspection Co., Ltd., Shenzhen, China). The measurements were repeated five times to get the average values.

The DSC analyses for the samples were performed on a TA Instruments Q20. The samples were first heated up to 200 °C, and then annealed for 5 min to eliminate thermal history, following by cooling down to 40 °C and then reheated up to 200 °C, at heating and cooling rates of 10 °C/min. Thermal gravimetric (TGA) analysis of the samples was performed by TA instrument Q50 from room temperature to 700 °C at a heating rate of 10 °C/min and a nitrogen gas flow of 90 mL/min. 

The specific surface area of CCB particles was tested using a BELSOBP-miniII automatic surface area Brunauer-Emmet-Teller (BET) analyzer (BELSOBP-miniII, MicrotracBEL, Osaka, Japan). The specific surface area was calculated according to the Brunauer–Emmett–Teller (BET) method.

To order to indicate intuitively the interface interactions between UHMWPE and CCB particles with various structures, the Raman mapping images of UHMWPE/h-CCB and UHMWPE/l-CCB composites were performed on a micro-laser confocal Raman spectrometer (Thermo Scientific DXR2xi, Waltham, MA, USA). The test condition of Raman was that the laser power is 1.5 mW, and the total exposure is 30 times with 0.025 s exposure time for each spectrum, Raman mapping images with an area of 50 μm^2^ in step sizes of 1 m. The imaging data is fitted through the Raman software “OMNICxi”, which gives each point a corresponding Raman characteristic peak spectrum.

## 3. Results

### 3.1. Characterization of h-CCB, and l-CCB Nanoparticles

We investigated the microstructure and properties of two kinds of CCB nanoparticles. These particles are often agglomerated into large aggregates due to the strong Van der Waals force between particles, leading to a high-percolation threshold value (15–20 wt%) for achieving the interconnected conductive networks [37]. This is mainly due to the zero-dimensional shape of the filler, which is hard to construct highly efficient conductive networks [38]. Figure 2 depicts the morphology and specific surface area of two kinds of nanoscale h-CCB or l-CCB particles. In Figure 2a,b, compared to l-CCB materials, we observed that the particle size of h-CCB is smaller than l-CCB, and most of h-CCB particles aggregate each other, it shows that h-CCB possesses a high specific surface area and strong polar interaction between particles. Figure 2c presents the N_2_ adsorption/desorption isotherms of two kinds of different specific surface area CCB nanoparticles; it suggests that the large specific surface area of h-CCB particles was attributed to its abundant porous structure, and the abundant mesoporous structure and high specific surface area of h-CCB particles enable the combined forces of numerous Van der Waals forces to become much stronger, which is beneficial for forming the strong interface interactions of mechanical interlocking between h-CCB and UHMWPE. Figure 2d indicates that Raman spectrum of h-CCB and l-CCB particles is mainly composed of D and G peaks at 1348 cm^−1^ and 1587 cm^−1^, in which I_D_/I_G_ value presented the level of the different graphitization and defects structure. Moreover, h-CCB particles present an evident 2G peak value in the Raman spectrum, the I_D_/I_G_ value of h-CCB is lower than l-CCB particles, implying the graphitization degree of h-CCB particles is higher than l-CCB particles. These results suggested that h-CCB particles with high graphitization level serve as conducting fillers in CPCs at low filler content, which has more competitive value in industry applications, owing to its outstanding electrical conductivity.

### 3.2. Microstructure of UHMWPE/CCB Composites

To vitally confirm the various dispersion and distribution characteristics of CCB particles, we perform optical microscopy (OM) imaging analysis. Figure 3 shows the cross-sectional optical microscopy of UHMWPE/l-CCB composites and UHMWPE/h-CCB composites. The obvious bright parts (UHMWPE region) and dark region (h-CCB or l-CCB rich region) that are connected can be observed clearly, indicating the formation of a segregated h-CCB or l-CCB network. In Figure 3a–c, with the increasing l-CCB content, some conductive networks are gradually constructed. Likewise, this phenomenon also appears in h-CCB filled UHMWPE matrix, as shown in Figure 3e,f. Compared to the loosely stacked l-CCB network, while the strong Van der Waals forces of h-CCB particle are easy to aggregate each other, and gradually form a denser and denser developed conductive network with increasing h-CCB concentrations. This may be attributed to the strong mechanical interlocked interactions of h-CCB particles and UHMWPE matrix, and the Brownian motion of CCB particles in the UHMWPE melt creates the distinctly interfacial transition layer and strongly interfacial adhesion, forming the densely stacked conductive network [35]. To sum up, the optical microscopy observations intuitively confirm the formation of high-quality segregated conductive networks. An interfacial transition layer with the small interface defect is achieved in UHMWPE/h-CCB composites due to the strong interfacial effects. Moreover, we observed that a compacted and oriented conductive pathway near the UHMWPE granules interfacial regions is formed at the optical microscopy of UHMWPE/h-CCB composites (Figure 3e). This result is likely due to the high viscosity of the UHMWPE matrix, which promotes the migration of h-CCB particles during the hot-compression process. With the increase of h-CCB particles dosage, the migration of h-CCB particles attached to the UHMWPE granules interfacial regions become dense and thick, which further reveals good interfacial adhesion between h-CCB and UHMWPE.

Using a reasonable approach of revealing perfect interfacial bonding in between polymer domains is very important. Figure 4 displays the interfacial microstructure of UHMWPE/CCB composites. The cryo-fractured surface of UHMWPE/CCB composites appears obviously the segregated h-CCB or l-CCB particles located at the interface between adjacent UHMWPE granules. Moreover, the cryo-fractured surface morphology confirms that there are no obvious agglomerates and clusters across the whole cryo-fractured surfaces of two UHMWPE/CCB composites with higher CCB loading. In Figure 4a–c, the micrographs revealed that the l-CCB dispersion and distribution morphology in UHMWPE/l-CCB composites with different l-CCB content is a relatively sparse morphology, the fractured surfaces become uneven, and some interfacial defects appear in the composites with the higher l-CCB loading. In Figure 4a, the red circle indicated the recognized CCB particles, and the phenomena of the selective dispersion and distribution at the interface can be detected by a higher-magnification SEM image. With the increase of the addition of l-CCB, from Figure 4c it can observed that l-CCB conductive pathways between UHMWPE particles begin a loosely conductive networks at 3 wt% 1-CCB concentration. With the CCB content increasing continually, many distinct interfacial voids appear in the cryo-fractured surfaces of UHMWPE composites with high l-CCB concentration. We can obviously observe some cracks of the composites with 5, 7, and 10 wt% l-CCB concentrations at higher magnifications, and two-phase interface of such composite is clear, which suggests the weak interfacial adhesion and compatibilization (Appendix A). This is attributed to the low specific surface area, poor polarity and physical adsorption ability of l-CCB particles. By contrast, UHMWPE/h-CCB composite shows a unique dense segregated structure that h-CCB are orderly distributed at the interfaces of UHMWPE granules. Moreover, the nanosized h-CCB particles are easily decorated onto UHMWPE granules due to their large specific surface area during the mechanical mixing, and then the rough UHMWPE granules experience the plastic deformation under hot-pressing stress, and constrain UHMWPE molecular chains diffusion in between granules. This synergistic interface effect forms a continuous and dense segregated structure in the process of compression molding. In Figure 4d–f, as the amount of h-CCB increases from 0.5 to 3.0 wt%, we observed that both h-CCB and l-CCB particles were orderly dispersed at the interfaces of UHMWPE granules, this result is attributed to the strong volume exclusion effect of the UHMWPE matrix with high viscosity. Moreover, the SEM morphology images of UHMWPE/l-CCB and UHMWPE/h-CCB with different filler content are very similar, and no voids and interfacial defects are found. These results suggested that UHMWPE/l-CCB and UHMWPE/h-CCB composites have excellent mechanical properties at a low loading content. Meanwhile, beyond 3.0 wt% h-CCB content, a unique segregated h-CCB conductive network composed of the mixture of denser stacked h-CCB particles bridging UHMWPE matrix is observed, as seen in Appendix A at higher magnifications. The evolution of various microstructures two kinds of UHMWPE/CCB composites is mainly attributed to the fabrication process of materials, and interface interactions of CCB-CCB and CCB-polymer matrix. In particular, different filler-matrix interfacial behavior was observed in the composites with CCB particles. In the composite with l-CCB particles, obvious gaps can be found, suggesting that the interactions between phases are relatively poor. By contrast, whereas the incorporation of h-CCB weight fraction into UHMWPE beyond 3 wt%, a compact segregated conductive network between the UHMWPE boundary regions is constructed. Furthermore, the low ratios of h-CCB filled UHMWPE matrix not only greatly guarantees the mechanical properties of the composites, but also realizes the construction of the segregated structure at low conductive percolation threshold values of 0.5~1 wt%.

Raman spectroscopy is a fast and useful tool for the characterization of interface structure for carbon-filled polymer composites. To explore the orderly dispersion and distribution of carbon fillers in a polymer matrix, we analyze the interface transition layer structure of h-CCB and l-CCB filled UHMWPE composite via using Raman mapping measurements. Figure 5 shows the polymer-filler interface structure of Raman mapping surface morphology of different UHMWPE/CCB composites. As the CCB content increases from 1.0 wt% to 3.0 wt%, we observed that the distribution of h-CCB or l-CCB significantly affects the interfacial layer microstructure of composite, owing to the formation of the different CCB networks. Moreover, the change of images color from red to blue represents of Raman signal of the UHMWPE matrix phase and CCB particles, respectively. The middle green color transition region is the UHMWPE-CCB co-existed phase, which becomes more and more distinct with the increase of h-CCB content, and in which a distinctly interfacial bonding layer is found. We also found that the segregated h-CCB lamella structures appear a compact segregated structure and interfacial bonding layer with different thicknesses, its h-CCB lamella thickness mainly depends on the content of h-CCB, as observed on the cross-sectional surface of UHMWPE/h-CCB composites as shown in Figure 5c,d. In Figure 5c,d, as the h-CCB content increases from 1 to 3 wt%, the thickness of the compact segregated h-CCB lamella increase from 4.44 µm to 22.65 µm, while the interfacial bonding layer is slightly enhanced, but the increasing trend is no obvious. This implied the strong interfacial interactions of h-CCB and UHMWPE are very helpful for the interfacial structure and properties of the composites. To achieve deeper insights into the different interfacial effects of CCB-filled composites, we analyzed that the presence of different colors represents the different position of Raman spectrum in detail in Raman mapping images of UHMWPE/l-CCB_0.5_, and UHMWPE/l-CCB_0.5_ composites, as is shown in Appendix A. Appendix A displays the corresponding Raman spectrum of Raman mapping images of UHMWPE/l-CCB_0.5_, and UHMWPE/l-CCB_0.5_ composites at the position of UHMWPE matrix materials (A curve), UHMWPE-CCB interfaces (B curve) and CCB particles (C curve). We found that the h-CCB-UHMWPE interfacial structure significantly constrains UHMWPE molecular chains micromovement compared to l-CCB particles, thereby leading to the conformational change of UHMWPE molecular chains in UHMWPE/h-CCB composites. This is related to the existence of the different interface bridging effects of l-CCB-UHMWPE, h-CCB-UHMWPE systems with the incorporation of CCB particles, which is consistent with SEM results. More importantly, the Raman spectrum of UHMWPE/h-CCB composite significantly presents an overlapped characteristic peak at 1348 cm^−1^ and 1587 cm^−1^. In Appendix A, from this characteristic peak shift change of Raman spectrum of composites, the crystallinity of UHMWPE/h-CCB composite decreases with increasing h-CCB content, we speculated that the addition of h-CCB particles is likely to impact their conformational and crystalline structure of UHMWPE matrices. This further suggested that plentiful h-CCB nanoparticles as an enhanced transition layer is embedded into the UHMWPE matrices, which is contributed to the retention of the mechanical properties of high-loading polymeric-based composites. 

DSC is one of the most effective methods for characterizing the effect of inorganic CCB in polymer matrices on the crystallization performance and the interaction of filler/polymer under the process of the crystallization of the crystal growth and nucleation. Therefore, further discussion on the thermal property parameters regarding the melting point (T_m_), enthalpy of fusion (ΔH_m_) and crystallinity (χ_c_) are necessary. Appendix A shows the DSC curves of pure UHMWPE, UHMWPE/l-CCB and UHMWPE/h-CCB composites. The detailed information of these DSC curves is listed in Appendix A. In Appendix A, the crystallinity (χ_c_) of UHMWPE gradually decreases with an increase of CCB content, and the reason for the decrease is mainly ascribed to the extremely high viscosity of the UHMWPE matrix; rigid CCB particles are segregated and evenly dispersed in UHMWPE boundary regions due to the volume exclusion effect of the matrix, limiting the alignment of UHMWPE chains and decreasing the melting points (T_m_) of the obtained composites, which is consistent with the reported results of the crystallization properties of high specific surface area MWCNT/UHMWPE [36], large aspect ratio graphene nanoplatelets/UHMWPE [39] and multi-scale mesoporous structural bamboo charcoal/UHMWPE composites [40]. This result further confirms that the crystal structure of UHMWPE is influenced by the h-CCB content—that is, physically interlocking interactions of high polar CCB particles impede the movement of UHMWPE molecular chains, which is also consistent with Raman mapping results. In Appendix A, the melting curves of DSC find only one melting peak corresponding to that of UHMWPE, which indicates the incorporation of CCB does not affect the crystalline structure of UHMWPE. Besides, the obtained results from the cooling curves Appendix A show the initial crystallization temperature (T_c, onset_) value of the crystallization temperature of UHMWPE is 121.16 °C, while is no obvious change compared to UHMWPE/CCB composites. For instance, when CCB content is 5 wt%, the Tc, onset of h-CCB or the l-CCB modified UHMWPE composites is 120.85 and 120.46 °C, respectively, which is close to the initial crystallization temperature of pure UHMWPE (121.16 °C).

To obtain a better understanding the impact of the interfacial interactions of the addition of various CCB into UHMWPE matrix on the thermal stabilities of the composites, we further evaluate the incorporation of h-CCB and l-CCB content on the thermal stabilities of the composites via using TGA analysis, separately. Appendix A depicts pure UHMWPE, UHMWPE/h-CCB and UHMWPE/l-CCB composites of TGA curves and DTG curves. The relevant thermal data, including the temperature of 5, 30, 50% weight loss (T_5_, T_30_, and T_50_) of the composites, the temperature at maximum weight loss rate (T_max_), heat resistance index (HRI) and charred residue quality are summarized in Appendix A. In Appendix A, the initial decomposition temperature of UHMWPE/h-CCB and UHMWPE/l-CCB composites with 0.5 wt% CCB concentration has been improved compared to pure UHMWPE, which is due to the interfacial bonding between polar CCB particles and UHMWPE granules, effectively hindering the transformation of heat in UHMWPE molecular chains, obviously enhancing their thermal stabilities. This result is consistent with the microstructure of the composite characterized by SEM. Moreover, the initial decomposition temperature of UHMWPE/h-CCB composites significantly increases as the h-CCB content increases compared to pure UHMWPE and UHMWPE/l-CCB, the T_5_, T_30_ and T_50_ of UHMWPE/h-CCB_10_ composites are increased by 9.67, 11.5 and 13.19 °C, respectively. This is attributed to high-quality CCB network structures that play a thermal barrier role, restrict the heat transfer in the UHMWPE matrix and simultaneously delay the thermal decomposition progress of composites as well. Furthermore, the T_5_, T_30_, T_50_ and heat resistance index (HRI) of UHMWPE/h-CCB composites is greater than UHMWPE/l-CCB composites, and these results further illustrate that h-CCB particles are conducive to enhance the interface effect and thermal stability properties of composites.

### 3.3. Mechanical Properties

The mechanical properties of most polymer composites rely on various factors, including polymer/filler type, method of preparation, dispersion or distribution of fillers [41] and polymer-filler interfacial interaction [42]. In addition, the excellent mechanical properties play an important role in the industrial practical applications of shielding materials as well. It is well-known that the mechanical performance is predominately determined by the interfacial adhesion of the adjacent polymer domains in CPCs materials [43]. Figure 6 shows the mechanical properties of pure UHMWPE, UHMWPE/h-CCB and UHMWPE/l-CCB composites. In Figure 6, when CCB content is 1 wt%, it can be seen that UHMWPE/CCB composites show higher mechanical properties compared to pure UHMWPE, due to the incorporation of good-dispersion CCB particles into UHMWPE matrices as nucleating agents [24], resulting in the stable conformation of the UHMWPE macular chains. Moreover, high loading UHMWPE/h-CCB composites greatly maintain mechanical performance compared to the same addition of UHMWPE/l-CCB composites, the elongation at break value of UHMWPE/h-CCB composites is also significantly higher than that of UHMWPE/l-CCB composites, further suggesting the interfacial adhesion of h-CCB nanoparticles is superior to the interaction of l-CCB, Furthermore, when the increase of h-CCB or l-CCB content to 10 wt%, the tensile strength and elongation at break value of UHMWPE/h-CCB composites are approximately two and six times that of UHMWPE/l-CCB composites, separately, as shown in Figure 6. The mechanical properties improvement in the strength and fracture of high-loading UHMWPE/h-CCB composite is thus attributed to the strong mechanical interlocking in between adjacent UHMWPE granules and h-CCB nanoparticles, good wetting interactions of h-CCB nanoparticles, leading to forming the dense and rigid h-CCB particle networks that can greatly transfer stress and dissipate energy [44]. Numerous matrix fibrils are formed on the tensile fractured surface of UHMWPE/h-CCB composite, and a typical ductile fracture is found. Appendix A further demonstrated that endowing the good interfacial adhesion between UHMWPE and h-CCB nanoparticles, making the UHMWPE/h-CCB composite more competitive as segregated conductive polymer materials, which can resist high stress in some specific fields. These results are consistent with the above-obtained SEM, OM, Raman mapping imaging and TGA characterization results. 

To further elucidate fracture and enhancement mechanism, the tensile fractured surfaces of UHMWPE/h-CCB, and UHMWPE/l-CCB composites, Figure 7 presents the broken tensile surfaces of l-CCB particles filled UHMWPE-based composite compared to a unique UHMWPE/h-CCB composite. The surface on the composite surface is rough because of the presence of h-CCB nanoparticles. No obvious h-CCB particles agglomeration can be observed in the composites, indicating that the dry-blending technique is an effective method to uniformly disperse the h-CCB particles in the polymer matrix. Figure 7a–c show that the surface of the composite material had some small voids, and some l-CCB particles are not well dispersed in the matrix, which was due to the poor interfacial mechanically interlocked in UHMWPE matrix, resulting in the decrease of mechanical properties of composites. Meanwhile, Figure 7d–f show that good interface quality is produced —that is, perfect contact is observed (no voids).

### 3.4. Electrical Conductivity

As one of the key parameters of the electromagnetic interference (EMI) shielding material, high electrical conductivity (σ) plays a vital role in EMI shielding application materials. Figure 8 shows the σ of UHMWPE/h-CCB and UHMWPE/l-CCB composites. In Figure 8a,b, the σ increases gradually with the increase of h-CCB or l-CCB loading weight ratios. Moreover, UHMWPE/h-CCB composites possess a higher σ than that of UHMWPE/l-CCB composites with the same concentration, which should be attributed to the improved conductive network of h-CCB particles. Besides, according to theoretical percolation threshold [45,46]: σ = σ_0_ (φ − φ_c_)^t^, for φ > φ_c_, where φ is the volume fraction of the fillers, φ_c_ is the volume percolation concentration, σ and σ_0_ represents the electrical conductivity of the composites and the conducting fillers, respectively. t represents a critical exponent reflecting the dimensionality of the system. To obtain the φ_c_ value, the linear fits of log σ vs. log (φ − φ_c_) is performed for each estimated “φ_c_”, as presented in the interior illustration of Figure 8b, and it is found that φ_c_ value of UHMWPE/h-CCB is far less than UHMWPE/l-CCB composite, and the percolation threshold of UHMWPE/h-CCB composite is 0.49 wt%. In comparison, we also found that the σ values of UHMWPE/l-CCB composites as a function of CCB content display a linear increasing trend, the estimated percolation threshold value of UHMWPE/l-CCB composite is more than 5 wt%. The φ_c_ value UHMWPE/h-CCB composite is far lower than that of UHMWPE/l-CCB composite. Furthermore, UHMWPE/h-CCB_10_ already gains a desirable electrical conductivity, and it is exciting that an excellent σ of UHMWPE/h-CCB composites (0.12 S/cm) with low h-CCB loading of 10 wt% far exceeds the target value (1.31 × 10^−2^ S/cm) for commercial use as EMI shielding materials, which can fulfill the commercial EMI shielding application. The achievement of outstanding σ values in UHMWPE/h-CCB_10_ can be correlated with the different CCB networks formed in the UHMWPE matrix. For UHMWPE/h-CCB_10_ composites, the formation of the interconnecting and dense network reveals that a large number of h-CCB are concentrated at the interfaces of UHMWPE domains and closely overlapped, which is more conducive to the transmission of electrons and interfacial polarity. Moreover, for UHMWPE/l-CCB_10_ composites, the σ values of UHMWPE/l-CCB_10_ composites mainly rely on l-CCB loading to realize higher σ owing to inherent l-CCB’s low electrical conductivity and poor interfacial energy characteristics. To further demonstrate the potential application of conductive UHMWPE/CCB composites, in Figure 8c,d a basic electronic setup based on a light-emitting diode (LED), together with various conductive UHMWPE/CCB composites, was assembled. The circuit measurement was carried out in UHMWPE/h-CCB_3_ composite and UHMWPE/l-CCB_7_ composite, and as expected, the LED bulbs can be glowed by the circuit, demonstrating the percolation behavior of UHMWPE/h-CCB segregated composite at a low percolation threshold, and have desirable σ value. Overall, utilizing an appropriate amount of CCB can not only realize high performance simultaneously, but also achieve the EMI materials with excellent EMI SE. 

### 3.5. Electromagnetic Interference (EMI) Shielding Performance

Figure 9a–f displays the EMI shielding performance and EMI shielding mechanism diagram of segregated UHMWPE/l-CCB and UHMWPE/h-CCB composites with different CCB content. In Figure 9a, it is clear that the EMI SE are continuously improved with increasing CCB filler loading due to the improvement in well-connected conductive network and electrical conductivity in the UHMWPE matrix. For instance, at 10 wt% h-CCB content, the EMI SE of the UHMWPE/h-CCB_10_ composite is found to be a satisfactory EMI SE of 22.3 dB at X band (8.2–12.4 GHz), and which already exceeded the requirement for commercial EMI SE (20 dB)—this signifies 99.9% blocking of the EM wave. Moreover, in Figure 9a, we found that the extent of increase in EMI SE corresponds to the extent of h-CCB incorporated in the UHMWPE matrix. The incorporation of h-CCB in UHMWPE matrix is the most significant contributor to the EMI SE of the composite compared to UHMWPE/l-CCB composite. To comprehend the EMI shielding mechanism for the UHMWPE/h-CCB composite composites, the contribution of absorption SE (SE_A_)and reflection SE (SE_R_) is disassembled from the total SE (SE_T_), and is shown in Figure 9b–d. The detailed SE_A_, SE_R_ and SE_T_ Values at 8.2 GHz are summarized in Appendix A. The relationship between power coefficients [i.e., reflection (R), absorption (A) and Transmission (T)] and frequency of the segregated UHMWPE/l-CCB_15_ and UHMWPE/h-CCB_15_ composites is illustrated in Appendix A. For UHMWPE/h-CCB_15_ composite, the R value raises to 0.78, and the T value keeps at quite a low level. A gradually decreasing trend is observed for A values caused by the huge increase of R. Meanwhile, the UHMWPE/l-CCB_15_ shows a low R value of 0.49 at 8.2 GHz. Moreover, from Figure 9, it could be significantly observed that SE_A_ contributes overwhelmingly to SE_T_, and this result suggests that absorption is the dominant shielding mechanism. Furthermore, we use the mechanism diagram to vividly show the entire process of electromagnetic waves passing through the composites, as shown in Figure 9e,f. The main way for UHMWPE/h-CCB composites to shield EM waves is absorption. When the electromagnetic wave reaches the surface of the composites, part of the electromagnetic wave is reflected, owing to the impedance mismatch. According to the impedance mismatch theory, the EMI SE of the segregated CPCs crucially depends on its high electrical loss and interfacial polarization loss. The desirable EMI shielding performance of CPCs often occurs synchronously with high electrical conductivity. The strong interfacial interaction and compatibility between h-CCB–h-CCB, h-CCB–UHMWPE is usually beneficial to forming more continuous conductive paths and subtle interface polarization, leading to the low percolation threshold and high EMI SE. Moreover, the obtained plentiful interconnecting conductive paths can impact the electrical conductivity of composite and decrease the separation distance between conductive fillers to a great extent. Overall, the high EMI SE of the segregated UHMWPE/CCB composites is attributed to such a dense conductive network with the fine interfacial bonding interactions—such interactions generate electrical loss and polarization loss, leading to the strong absorption and multiple scattering of incident EM waves.

## 4. Conclusions

UHMWPE/h-CCB and UHMWPE/l-CCB composites were prepared using simple and effective high-speed mechanical mixing combined with the compression molding method. The mechanical properties of high-loading UHMWPE/h-CCB composite were maintained greatly, and much higher than that of UHMWPE/l-CCB composite, revealing the strong interfacial interactions of UHMWPE matrix and h-CCB particles. The electrical threshold mechanism reveals that a low percolation threshold (0.49 wt%) is obtained in the UHMWPE/h-CCB composite due to the formation of a densely compacted h-CCB network, which is far lower than that of the percolation threshold of UHMWPE/l-CCB. UHMWPE/h-CCB exhibits an evident improvement of electromagnetic interference shielding effectiveness (EMI SE) when the dosage of h-CCB exceeds its electrical percolation threshold, the UHMWPE/h-CCB composite exhibits a satisfactory EMI SE value of 22.3 dB at 10 wt% h-CCB content, which is as four times that of UHMWPE/l-CCB composite with same l-CCB dosage (5.6 dB). Moreover, the tensile strength and elongation at break value of UHMWPE/h-CCB_10_ composites are approximately two and six times that of UHMWPE/l-CCB_10_ composites. In a word, the addition of h-CCB particles not only effectively increases the thermal stability of segregated conducting UHMWPE-based composite, but also ensures the excellent mechanical properties and acceptable EMI SE in practical applications. This low-cost and high-performance segregated conductive polymer composite is a promising candidate for EMI shielding and antistatic fields, which will expect to open the door for next-generation cost-effective EMI shielding materials to cater academic and industrial applications in the future.

## Figures and Tables

**Figure 1 nanomaterials-11-02074-f001:**
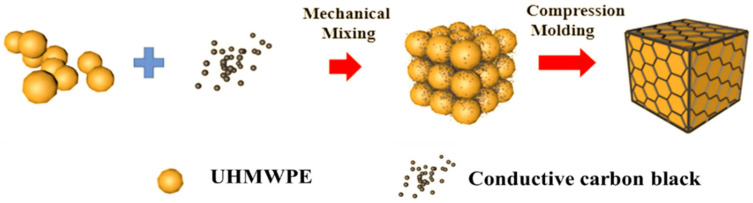
Preparation schematic illustration of segregated UHMWPE/CCB composites.

**Figure 2 nanomaterials-11-02074-f002:**
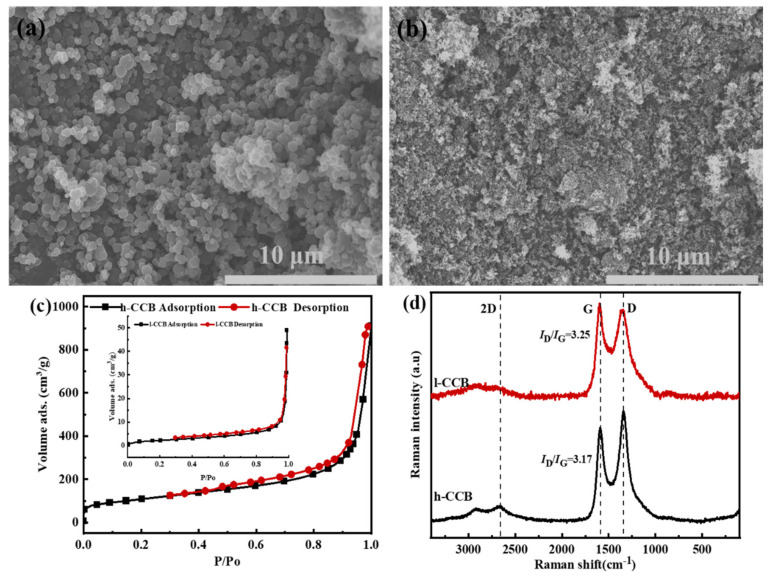
SEM images of CCB particles: (**a**) l-CCB particle; (**b**) h-CCB particles; (**c**) N_2_ adsorption/desorption isotherms, (**d**) Raman spectroscopy of nanoscale h-CCB and l-CCB particles.

**Figure 3 nanomaterials-11-02074-f003:**
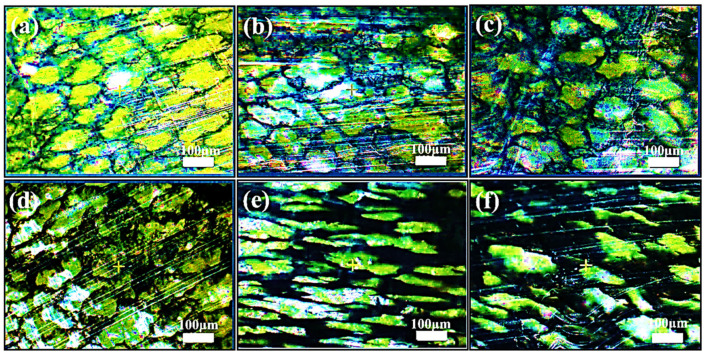
Optical microscopy images of the cross-sectional surface of UHMWPE/CCB composites: (**a**) UHMWPE/1-CCB_0.5_, (**b**)UHMWPE/1-CCB_1_, (**c**)UHMWPE/l-CCB_3_, (**d**)UHMWPE/h-CCB_0.5_, (**e**)UHMWPE/h-CCB_1_ and (**f**)UHMWPE/h-CCB_3_.

**Figure 4 nanomaterials-11-02074-f004:**
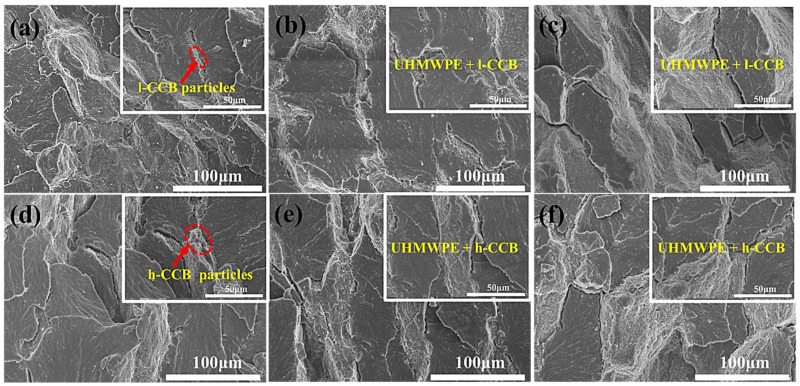
FE-SEM morphology of the cryo-fractured surface of UHMWPE/CCB composites: (**a**) UHMWPE/1-CCB_0.5_; (**b**) UHMWPE/1-CCB_1_; (**c**) UHMWPE/1-CCB_3_; (**d**) UHMWPE/h-CCB_0.5_; (**e**) UHMWPE/h-CCB_1_; and (**f**) UHMWPE/h-CCB_3_.

**Figure 5 nanomaterials-11-02074-f005:**
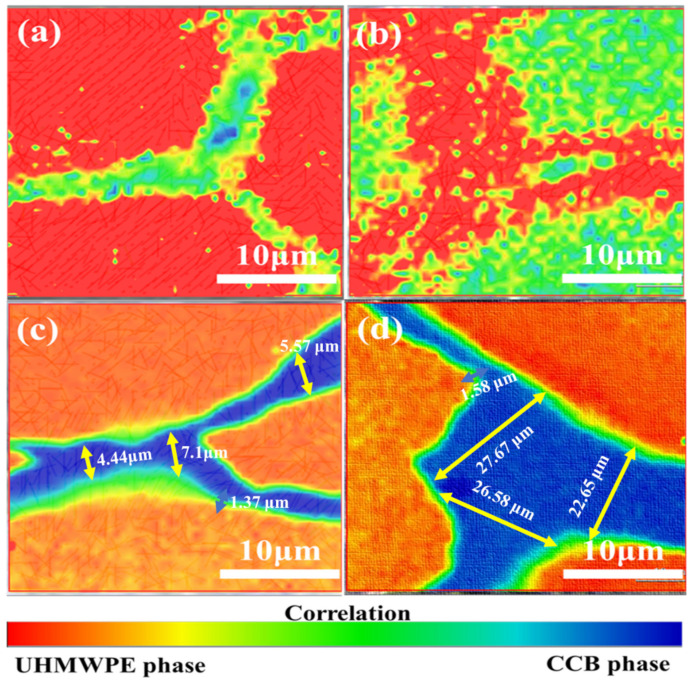
Raman mapping images of UHMWPE/CCB composites with different concentrations: (**a**) UHMWPE/1-CCB_1_; (**b**) UHMWPE/1-CCB_3_; (**c**) UHMWPE/h-CCB_1_; and (**d**) UHMWPE/h-CCB_3_.

**Figure 6 nanomaterials-11-02074-f006:**
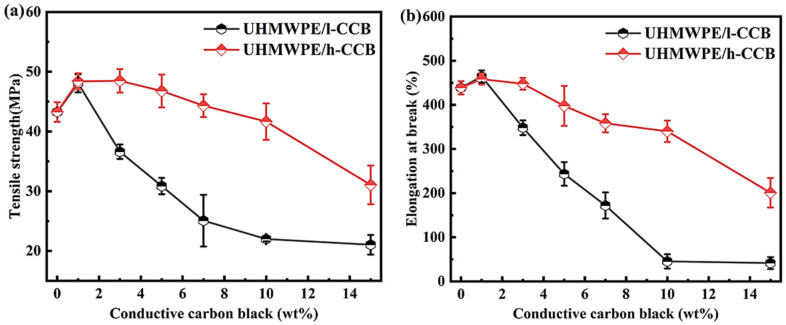
(**a**) tensile strength and (**b**) elongation at break of UHMWPE/CCB composites.

**Figure 7 nanomaterials-11-02074-f007:**
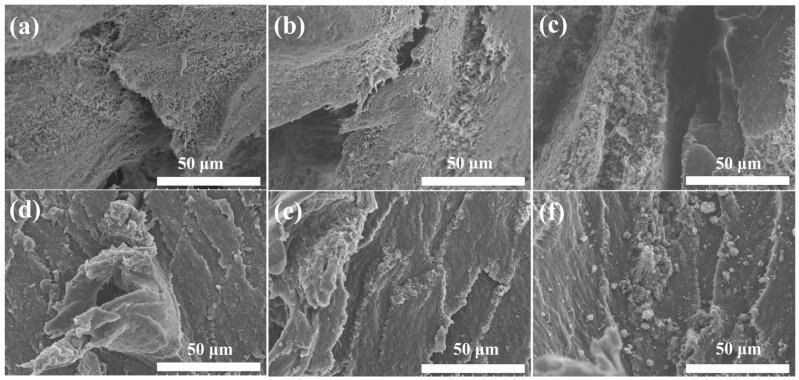
SEM images of the tensile fractured surfaces of UHMWPE/l-CCB composites: (**a**) UHMWPE/l-CCB_5_; (**b**) UHMWPE/l-CCB_10_; and (**c**) UHMWPE/l-CCB_15_, and UHMWPE/h-CCB composites: (**d**) UHMWPE/h-CCB_5_; (**e**) UHMWPE/h-CCB_10_; and (**f**) UHMWPE/h-CCB_15_.

**Figure 8 nanomaterials-11-02074-f008:**
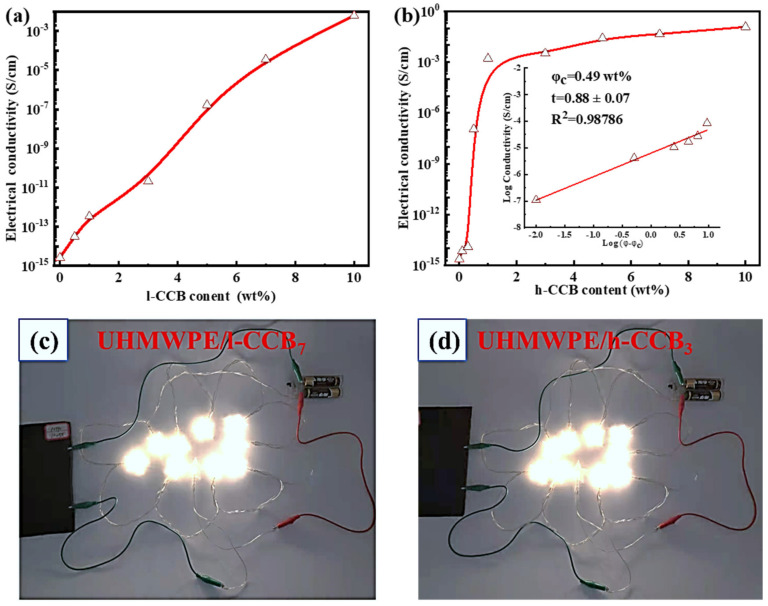
Comparation of σ of UHMWPE/CCB composites with the different CCB content: (**a**) UHMWPE/h-CCB; and (**b**) UHMWPE/l-CCB; the actual conductive circuit of the glowing LED bulbs energized by (**c**) UHMWPE/l-CCB_7_; and (**d**) UHMWPE/h-CCB_3_ composite.

**Figure 9 nanomaterials-11-02074-f009:**
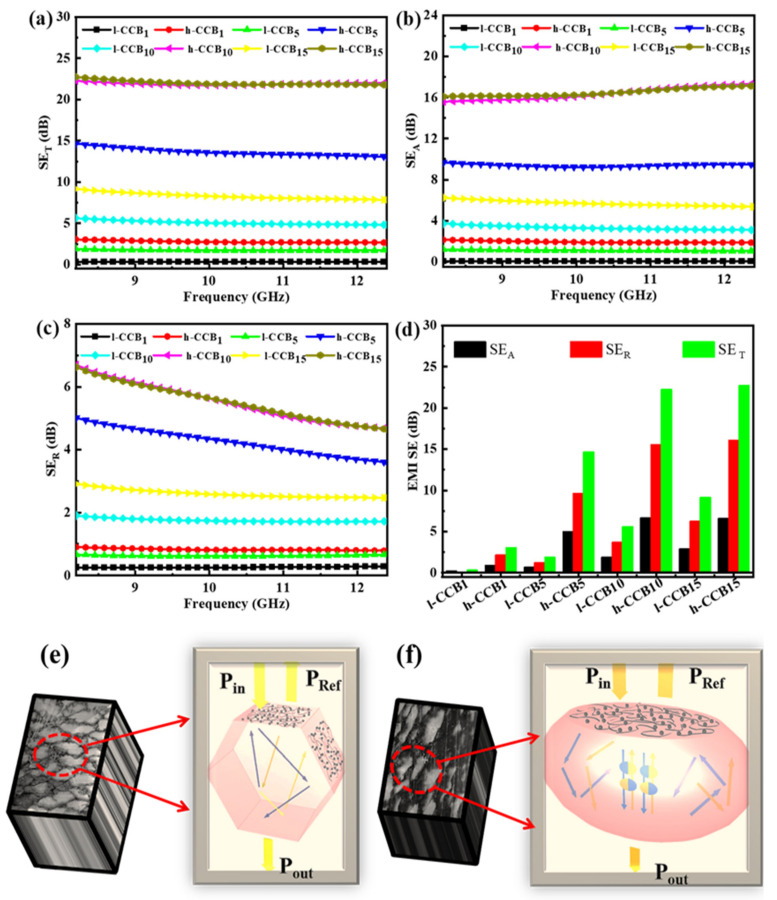
(**a**) SE_T_; (**b**) SE_A_; and (**c**) SE_R_ of UHMWPE/h-CCB and UHMWPE/l-CCB composites with different CCB contents; as well as (**d**) Comparison of EMI SE of UHMWPE/h-CCB and UHMWPE/l-CCB composites with different CCB weight ratios at the thickness of approximately 1.5 mm; (**e**) EMI shielding mechanism diagram of UHMWPE/l-CCB; and (**f**) UHMWPE/h-CCB composites.

## Data Availability

Data is contained within the article and Appendix A.

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
