# Peer review of "Endowing Acceptable Mechanical Properties of Segregated Conductive Polymer Composites with Enhanced Filler-Matrix Interfacial Interactions by Incorporating High Specific Surface Area Nanosized Carbon Black"

_nanomaterials, 2021, doi:10.3390/nano11082074_

Round 1

Reviewer 1 Report

The manuscript uses a facile method for fabricating ultra-high molecular weight polyethylene/conducting carbon black composites with the segregated structures. The interfacial interactions, mechanical properties, electrical properties, and electromagnetic interference shielding properties were investigated. The paper is well organized, the presented results are interesting, and the topic is relevant to this journal. However, several main scientific concerns are there. Thus, a major revision is required to address the following main concerns, before further evaluating the suitability for the publication of this work.

1 Title: The title “Desirable Mechanical Properties…by Incorporating Nanosized Carbon Black” should be reconsidered. From Fig. 6, the tensile strength and elongation at break of UHMWPE/CCB composites have not improved much, and the mechanical properties of most composites have even deteriorated. And from SEM, the Carbon Black did not achieve nanosized dispersion in composites, so the title is inappropriate.

2 Introduction: The authors mentioned that “The segregated conductive pathway has proven to outperform in enlarging conductive loss mechanism, particularly for polymer-based EMI shielding materials”. The topic background of the manuscript needs to be lighted further to broaden the impact, related literature about constructing segregated conductive pathway like: Composites Part A: Applied Science and Manufacturing (2021): 106376 is recommended to cite.

3 Results: “In Figure 4(d-f), as the amount of h-CCB increases from 0.5 to 3.0 wt%, h-CCB particles remain orderly dispersed at the interfaces of UHMWPE granules instead of penetrating their interior”, “And the h-CCB lamella structures appear a compact segregated structure with different thicknesses, Figure 4(f)”. These conclusions cannot be obtained from Fig. 4, the SEM images of UHMWPE/l-CCB and UHMWPE/h-CCB with different filler content are very similar, which cannot prove the conclusion obtained by the authors. The authors are suggested to focus more on analyzing the Raman mapping images at different magnification to get more reliable conclusions.

4 The unit of electrical conductivity is S/m in the context in section 3.4, but the unit of electrical conductivity is S/cm in Fig. 8. The unit should be consistent in the paper.

5. There are some relevant literature to be cited, such as Composites Part A: 2021, 145, 106376; Polymer, 2011, 52 (18), 4001-4010; Nanotechnology, 2013, 24 (12), 125704.

Reviewer 2 Report

This study presents a strategy for preparing the segregated conducting UHMWPE based composites with desirable mechanical properties. This is an important research field and the manuscript presents an interesting contribution for better understanding the effect of the inclusion of carbon black. English is of an acceptable quality. Only a minor key point:

  1. The fillers are conductive. Is it possible to compare the results against the composites with nonconductive fillers? It may elucidate the effect of the conductivity of the fillers.

A minor revision is recommended.

Author Response

We sincerely appreciate the Reviewers for these constructive remarks and useful suggestions. We addressed each comment below and highlighted where the changes were made in manuscript with blue color. We wish the revised manuscript could meet the requirements of Nanomaterials. The point-by-point responses and revisions are as follows.

Response to Reviewer 2 Comments

Point 1: The fillers are conductive. Is it possible to compare the results against the composites with nonconductive fillers? It may elucidate the effect of the conductivity of the fillers.

Response 1: We thank the Reviewer for the interesting questions. To address the concern of the Reviewer, we chose wasted rubber pyrolysis carbon black (RP-CB) particles with the similar specific surface area with h-CCB as nonconductive fillers, using to prepare UHMWPE/RP-CB composite materials. It is found that either h-CCB or l-CCB make the composites conductive. Whereas UHMWPE/RP-CB composite with 15 wt% RP-CB content possesses a low electrical conductivity. We are interested to find that the tensile strength of the obtained UHMWPE/RP-CB15 composite is 32.96±2.81 MPa, which is close to the prepared UHMWPE/h-CCB15 composite (31.06±3.25 MPa), as shown in Table S1. This result can elucidate that both conductive carbon and nonconductive carbon fillers have the difference in intrinsic conductivity and distribution of fillers. And the specific surface area, and surface energy of the carbon fillers play a significant role in the mechanical properties of the composites at high loading content. Meanwhile, we also found nonconductive carbon materials that have such a good maintenance of the mechanical properties of composite materials. Therefore, the high addition amount of nano-carbon materials has an excellent universality for such a good retention rate of composite materials. We will also deeply explore the effect of the conductivity of the fillers on the structure and properties of composites in our following work.

Table S1 comparison of electrical conductivity and tensile strength for UHMWPE/RP-CB15, UHMWPE/l-CCB15, and UHMWPE/h-CCB15 composites

Sample

Tensile strength (MPa)

UHMWPE/RP-CB15

32.96±2.81

UHMWPE/l-CCB15

20.06±0.94

UHMWPE/h-CCB15

31.06±3.25

Round 2

Reviewer 1 Report

The authors have addressed my concerns and the work can be accepted as it is now.